# Microwave and Ultrasound Pre-Treatments for Drying of the “Rocha” Pear: Impact on Phytochemical Parameters, Color Changes and Drying Kinetics

**DOI:** 10.3390/foods10040853

**Published:** 2021-04-14

**Authors:** Begüm Önal, Giuseppina Adiletta, Marisa Di Matteo, Paola Russo, Inês N. Ramos, Cristina L. M. Silva

**Affiliations:** 1Department of Industrial Engineering, University of Salerno, via Giovanni Paolo II, 132, 84084 Fisciano, Italy; bonal@unisa.it (B.Ö.), gadiletta@unisa.it (G.A.); mdimatteo@unisa.it (M.D.M.); 2Department of Chemical Engineering Materials Environment, Sapienza University of Rome, via Eudossiana, 18, 00184 Rome, Italy; paola.russo@uniroma1.it; 3Escola Superior de Biotecnologia, CBQF—Centro de Biotecnologia e Química Fina–Laboratório Associado, Universidade Católica Portuguesa, Rua Diogo Botelho 1327, 4169-005 Porto, Portugal

**Keywords:** “Rocha” pear, ultrasound, microwave, drying, quality characteristics, empirical models

## Abstract

The objective of this research was to evaluate the effect of drying temperature and innovative pre-treatments (i.e., microwave and ultrasound) on “Rocha” pear drying behavior and quality characteristics, such as color, total phenolic content and antioxidant activity. Experiments were carried out with pear slabs subjected to microwaves (2450 MHz, 539 W, 4 min, microwave oven) and ultrasounds (35 kHz, 10 min, in an ultrasonic bath) as well as control samples. The drying process was conducted in a tray dryer at three different temperatures (50, 55 and 60 °C) and a fixed air velocity of 0.75 m/s. Microwave technology resulted in a higher quality deterioration in dried pear samples compared to those of controls and ultrasound pre-treated samples. The combined application of ultrasound pre-treatment and the higher drying temperature of 60 °C was characterized by the lowest color changes (ΔE = 3.86 ± 0.23) and higher preservation of nutritional parameters (total phenolic content, TPC = 345.60 ± 8.99; and antioxidant activity, EC_50_ = 8.80 ± 0.34). The drying characteristics of pear fruits were also analyzed by taking into account empirical models, with the Page model presenting the best prediction of the drying behavior. In conclusion, ultrasound application is a promising technology to obtain healthy/nutritious dried “Rocha” pear snacks as dietary sources for consumers.

## 1. Introduction

The “Rocha” pear (*Pyrus communis* L.) is a traditional Portuguese cultivar classified with a Protected Designation of Origin (PDO) [1]. In addition to being a commercially important agricultural product throughout the world, the “Rocha” pear is characterized by its excellent organoleptic properties, nutritional quality and high resistance due to its initial rigidity [2,3,4]. Pear fruits are a rich source of vitamins and minerals, dietary fiber and phenolic substances. Therefore, pear consumption has received increasing interest due to its potential health-promoting benefits, in particular high antioxidant activity, as well as anti-inflammatory, anti-bacterial, anti-diabetic, uro-disinfectant, sedative and antipyretic properties [5,6,7,8]. The “Rocha” pear can be maintained up to 10 months under controlled atmosphere storage (2–3 kPa O_2_ + 0.5–0.7 kPa CO_2_ at −0.5 °C and 95% relative humidity) [9]. However, long-term storage under a controlled atmosphere (CA) may promote physiological deterioration such as internal browning disorders and superficial scald, influencing the overall aspects of the pear fruit and therefore consumer acceptability [3,10]. Nevertheless, post-harvest losses represent a major problem [11], not only from environmental but also from economic perspectives [12,13]. Hence, there is a need to develop alternative strategies to contribute to environmental and economic factors and sustain the nutritional and overall quality of agricultural products such as “Rocha” pears. In this context, the drying process is one of the most important preservation techniques to prolong shelf-life and availability, maintain the high-added-value components, valorize traditional food products, reduce transportation and storage costs and present new ways of consumption. As a consequence of the consumer’s lifestyle/behavior changes, in recent years, the food dehydration industry, including dried pears, has become an important market segment due to increased demand for healthy/nutritious ready-to-eat products (i.e., snack preparations, integral breakfast cereals, bakery/confectionery products and rehydrated functional fruit recipes). Among the numerous techniques, conventional hot air drying is widely used in the industry as it enables uniform processing conditions, is easy-to-use, and produces better-dried products with optimal drying conditions [14,15,16]. However, this drying technique may promote the partial degradation of nutritional compounds and undesirable quality changes of the final product such as color, appearance and structural properties. These properties extensively depend upon drying conditions (e.g., air temperature and velocity). In this sense, recent advances in using novel food processing technologies have received much attention to reduce various adverse changes and prepare final dried products with fresh-like characteristics.

Among the so-called emerging technologies, ultrasound waves are applied as a pre-treatment prior to convective drying of various fruit and vegetables to accelerate the drying process, reducing drying time [17,18], enhancing mass transfer phenomena [19,20], preserving functional components [21,22], inactivating enzymes [23] and improving rehydration characteristics [24,25]. There are two main phenomena induced by ultrasound that can affect solid-like materials: sponge effect and cavitation. The mechanical force and cavitation effects cause structural changes to the plant tissue such as the formation of micro-channels, which can be beneficial for water diffusion [20,23,26,27]. Much of the research on drying and quality characteristics by these authors have demonstrated that the application of ultrasound has different mechanisms and effects on various fruits.

Recently, utilization of microwave energy has drawn appreciable attention, as it is stated to enhance the drying process by lowering the processing time and operational costs [28,29]. Microwave technology penetrates the foodstuff quickly by heating it from inside to outside, resulting in an increase of the product temperature and consequently, rapid water evaporation [27,30]. Despite its benefits for drying efficiency, this technology has some major drawbacks regarding the dried product’s quality, including color deterioration, the formation of hot/cold spots due to non-uniform heating, nutritional losses, reduction of volatile aroma compounds and possible textural damage [27,28,31,32]. Additionally, the high penetration power of microwave energy affects heat transfer behavior and may lead to overheating agricultural products, depending on the food’s properties (nature, permittivity, water content), the geometry of the material, dielectric properties and oven design [27]. The application of microwave energy as a pre-treatment and its effects on the drying process and quality attributes have been addressed by a limited number of authors [31,33].

Obtaining an in-depth understanding of the drying process by means of models is a crucial tool for designing industrial drying systems and optimizing process conditions [34,35]. The most common empirical models are those of Newton, Henderson and Pabis, Page, Modified Page, two-term and Wang and Singh and have been utilized for various food materials including quince [36], cherry [37], grape [38], peach and strawberry [39] and persimmon [40] to describe several aspects of drying behavior These modeling approaches allow prediction and simulation of industrial drying systems, and help choose the most suitable operating conditions for the effective drying of fruits and vegetables.

However, no scientific work has been published concerning the effect of such kinds of pre-treatments (i.e., microwave and ultrasound applications) on drying behavior and quality characteristics of the “Rocha” pear. There is a need to study the application of emerging technologies to ensure process efficiency and product quality, satisfying the increased demand for healthy and nutritious snacks. Therefore, the focus of this work was to investigate the effects of microwave (2450 MHz, 539 W, 4 min, microwave oven) and ultrasound (35 kHz, 10 min, ultrasound bath) technologies as a pre-treatment on the convective air drying characteristics of “Rocha” pear slices, using drying temperatures of 50, 55 and 60 °C. Different empirical models were attempted to predict the drying process. Moreover, dried and raw “Rocha” pears quality attributes were assessed on water activity, color, total phenolic content and antioxidant activity.

## 2. Materials and Methods

### 2.1. Raw Material Preparation

“Rocha” pears (*Pyrus communis* L. ‘Rocha’) were acquired in a local market (Porto, Portugal) and stored in a refrigerator at 4 °C (BOSCH DUO SYSTEM, Bosch, Gerlingen, Germany) until drying or experimental analysis, up to a maximum of 2 days. Pears were selected taking into account sensory attributes such as freshness, uniform size, color and absence of mechanical damage and disease symptoms. Prior to drying, samples were washed with tap water, dried with absorbent paper and peeled. “Rocha” pears were sliced using a stainless steel slicer in order to maintain a uniform thickness of each sample. The average diameter and thickness of pear slices were 38 ± 0.12 mm and 6 ± 0.05 mm, respectively.

The average moisture content of raw pears, determined according to the AOAC official method [41] at 105 °C for 24 h, was found to be 6.42 kg water/kg d.m (86.52 ± 0.66%, w.b.).

### 2.2. Pre-Treatments Applied to the “Rocha” Pear

Three kinds of dried pear samples were compared in this study: (a) control (C) (without application of ultrasound or microwave pre-treatment), (b) ultrasound pre-treated (US) and (c) microwave pre-treated (MW) before drying.

Prior to the ultrasound application, portions of 40 g of pear slabs were hermetically packaged in eight polyethylene bags (VWR, 120 × 170 × 0.05 mm) (6 slabs per bag). Bags were subjected to vacuum packaging at 1 mPa during 1.2 s of welding using a vacuum packaging machine (A300/41/42, Multivac, Wolfertschwenden, Germany). The vacuum-packaged samples were used to avoid adverse effects of rinsing out substances contained in the pear fruits. Ultrasound waves, when passing through a liquid medium, lead to mechanical vibration of the liquid. If the liquid medium contains dissolved gas, which is the case under normal conditions, microbubbles can form, grow and violently collapse due to the action of the sound wave. This phenomenon is called “acoustic cavitation” Therefore, the use of vacuum packaging might decrease cavitation effects in the pear samples by preventing direct contact with a liquid medium. This methodology is a key point before ultrasound application to food products, as recommended by Nowacka and Wedzik [24]. The pear samples were placed in a metal net at the bottom of an ultrasonic bath (Bandelin Sonorex RK 255H; 300 mm (L) × 150 mm (W) × 150 mm (H)) containing 3 L of distilled water and without any mechanical agitation. The ultrasound frequency was 35 kHz, and the nominal power of the ultrasounds was 160 W. The experiments were conducted for an ultrasonic processing time of 10 min at a temperature of 30 °C. The ultrasonic processing time and temperature conditions were chosen according to the literature [17,21,24,42]. During the ultrasound applications, the temperature increase was lower than 2 °C. The slabs of each treatment were removed from the ultrasound bath, drained with absorbent paper and then subjected to the drying process.

10 pear slabs (approximately 100 g) were placed in a microwave oven (Beko 20 L, P.C.R, dimensions: 454 mm (W) × 330 mm (D) × 262 mm (H)) with a rotating plate and then subjected to microwave energy at a frequency of 2450 MHz and power of 539 W for 4 min. [33]. The total fresh mass of pear slabs was approximately 300 g, and after microwave heating, the weight of the samples was 251.8 ± 0.6 g, therefore evaporating around 50 g of water. The energy per mass was around 1294 J/g. Each run was repeated three times to prepare 30 pear slabs for each set. After microwave pre-treatment, all samples were cooled down and then subjected to drying.

### 2.3. Drying Experiments

Drying experiments of the control (C), microwave (MW) and ultrasound (US) pre-treated pear slabs were conducted in a pilot plant convective air tray drier with forced air (Armfield UOP8, Ringwood, England) and controlled air temperature and velocity, also including a metallic tube that allows air recirculation [43]. On-line acquisition of weight loss during the drying was recorded every 3 min by means of a digital balance (Sartorius, Goettingen, Germany) attached to the drying equipment and connected to a computer (Hewlett–Packard Vectra, Palo Alto, CA, USA). The tray dryer was previously heated to the set point temperature, and drying conditions were stabilized and maintained for at least 30 min before sample drying. An amount of approximately 300 g pear samples was divided into three similar portions and loaded onto the dryer’s three top trays. The drying processes of the pears were conducted until a final constant moisture content of less than 0.1 ± 0.01 kg water/kg d.m., using air drying temperatures of 50, 55 and 60 °C at a fixed air velocity of 0.75 m/s. The air temperature was monitored with a squirrel data-logger (Grant Instruments 1023, Cambridge, England) and thermocouple wires connected to each tray. The air velocity was measured regularly by a vane anemometer (Airflow LCA 6000, Buckinghamshire, England). Outlet dry and wet bulb air temperatures were recorded to calculate its relative humidity values through a psychrometric chart calculator [44]. Average air relative humidity values were 33.97 ± 0.81%, 31.67 ± 0.46% and 27.27 ± 0.75%, respectively, for drying temperatures of 50, 55 and 60 °C.

Drying experiments were performed at each temperature in three independent sets (C, MW and US), each with three replicates. After drying, pear samples were cooled down to room temperature, further packaged using a vacuum packaging machine and allowed to stand in the dark until further analysis.

### 2.4. Modeling of Drying Kinetics

“Rocha” pears’ drying kinetics was expressed as a moisture ratio (XR) as a function of time:(1)$$\mathrm{XR}=\frac{X-X_{e}}{X_{0}-X_{e}}$$
where X, X_0_ and X_e_ were, respectively, the moisture contents at a given time, initial and equilibrium (kg water kg dry matter^−1^). This equation can be simplified because X_e_ is small compared to X and X_0_, and the error of considering it negligible is not significant [39,45]. Therefore, X_e_ was set equal to zero in all drying models. Moreover, this assumption was confirmed by calculating an average relative humidity value for all experiments and the corresponding X_e_ using sorption isotherms.

Several empirical models commonly applied to fruit materials were analyzed to fit “Rocha” pear drying data including the Newton model [46,47]:(2)$$\frac{X-{\;X}_{e}}{X_{0}-{\;X}_{e}}=\exp\left(-\mathrm{kt}\right)$$
the Henderson & Pabis model [48,49]:(3)$$\frac{X-{\;X}_{e}}{X_{0}-{\;X}_{e}}=a\;\exp\left(-k\;t\right)$$
and the Page model [50]:(4)$$\frac{X-{\;X}_{e}}{X_{0}-{\;X}_{e}}=\exp(-{k\;t}^{N})$$

Drying data were normalized to each pear sample’s initial moisture content to allow comparison between different pre-treatment effects. A non-linear regression analysis was used to estimate the coefficients (drying rate constant, k (min^−1^ or min^−N^), and drying coefficients, a and N) of the tested models. For each model, the goodness of fit analysis was based on the following statistical parameter values: coefficient of determination (R^2^) and the standard deviation of the experimental error (s). The model better describing each pear sample’s drying characteristics was chosen according to a higher coefficient of determination (R^2^) and a lower standard deviation of the experimental error (s).

### 2.5. Quality Parameters Evaluation

#### 2.5.1. Water Activity

Each pear sample’s water activity was determined using a water activity meter (Aqualab, Model Series 3 TE, Decagon Devices, Pullman, WA, USA) at 23 °C, previously calibrated with a standard solution of water activity of 0.760. Fresh and dried pears were inserted in a converter chamber, and a_w_ values were recorded after equilibration. Each measurement was carried out in triplicate and the mean values calculated.

#### 2.5.2. Color Properties

The color parameters of fresh and dried pear slabs (control and microwave and ultrasound pre-treated) were evaluated using a Minolta CR-400 colorimeter (Konica-Minolta, Osaka, Japan) based on the CIELAB coordinate system. Before the measurements, the instrument was calibrated with a white ceramic plate. Three readings of three different replications were conducted for each sample.

Three color parameters (L*, a* and b*) were recorded. The L* value represented the brightness and varied from 0 (black) to 100 (white). The chromaticity coordinates indicated for a* the redness (+)/greenness (−) and for b* the yellowness (+)/blueness (−). The whiteness index (WI) and total color difference (ΔE) were derived from the values of L*, a* and b*. These values were crucial to obtaining an in-depth understanding of the impacts of pre-treatments and drying process conditions (i.e., temperature and time) on color changes.

The whiteness index (WI) represented the overall whiteness of fruits and vegetables that may indicate the extent of discoloration during the drying process [51]. WI was used to express the degree of whiteness of pear samples, and was calculated as follows [38]:(5)$$\mathrm{WI}=100-\;\sqrt{{\left(100-L^{*}\right)}^{2}+\left(a^{*}{}^{2}\right)+\left(b^{*}{}^{2}\right)}$$

Total color difference *(*∆E) represented the degree of overall color changes of pear samples and was determined according to Equation (6) [52]. The reference value for total color difference (∆E) was the fresh “Rocha” pear.
(6)$$∆E=\;\sqrt[{2}]{{\left(L^{*}-L_{0}^{*}\right)}^{2}+{\left(a^{*}-a_{0}^{*}\right)}^{2}+{\left(b^{*}-b_{0}^{*}\right)}^{2}},$$
where $L_{0}^{*}$, $a_{0}^{*}{\;\mathrm{and}\;b}_{0}^{*}$ were the color parameters for the fresh pear and $L^{*}$, $a^{*}$ and $b^{*}\;$the corresponding values after drying.

#### 2.5.3. Sample Extraction

The extraction methodology used was adapted from Salta et al. [53] and Önal et al. [54] with some modifications. The samples’ extracts for the determination of total phenolics and total antioxidant capacity were prepared by homogenizing 5 g (±0.01) fresh sample or 2 g (±0.01) dried sample in 30 mL and 35 mL of methanol, respectively (100%, *v*/*v*) (CHROMASOLV^®^, for HPLC, ≥99.9%, Sigma-Aldrich, St. Louis, MO, USA), throughout an Ultra-turrax (Ika digital T25, IKA^®^-Werke GmbH & Co. KG, Staufen, Germany) at 11 rpm. The obtained mixture for each sample was stirred for 10 min by a vortex, and then extracts were filtered throughout a Whatman No:2 filter paper. Three extracts were prepared for each sample, and measurements were performed for each replicate.

#### 2.5.4. Total Phenolic Content

The total phenolic content in pear samples was determined in accordance with the Folin-Ciocalteu colorimetric method previously reported by Önal et al. [54]. Describing briefly, the pear extracts were oxidized by the Folin-Ciocalteu reagent in the mixture of sodium carbonate solution and distilled water. The reaction mixture was allowed to stand at room temperature for 1 h, and then the absorbance was measured, in triplicate, at 760 nm, using a visible spectrophotometer (Novaspec II, Piscataway, NJ, USA). The calibration curve was constructed with standard Gallic acid, and results were reported as milligrams of gallic acid equivalents per 100 g of dry basis (mg of GAE/100 g d.m.).

#### 2.5.5. DPPH Radical Scavenging Activity

The DPPH (1,1-diphenyl-2-picrylhydrazyl) radical scavenging assay was used to assess the antioxidant activity of fresh and dried pears according to the method proposed by Adiletta et al. [55] and Önal et al. [54]. Various volumes of pear extracts were prepared and then mixed with 3.5 mL of 6 × 10^−5^ m of DPPH methanol solution in cuvettes. The reaction mixtures were vortexed for 30 s. After 30 min incubating in the dark and at room temperature, absorbance values were recorded at 517 nm using a visible spectrophotometer (Novaspec II, Piscataway, NJ, USA). Observed color variation from purple to yellow demonstrated the progress of the reaction due to the reduction of the 1,1-diphenyl-2-picrylhydrazyl (DPPH) solution complex. Pure methanol (*v*/*v*) was used as the blank, and the preparation of the control sample was without adding any pear extract.

The total antioxidant activity of pear extracts was expressed as the percentage inhibition of the DPPH radical and calculated [56,57] as:(7)$$\;\%\;\mathrm{Antioxidant}\;\mathrm{Activity}=\frac{\mathrm{Abscontrol}\;-\;\mathrm{Abssample}}{\mathrm{Abscontrol}}\;\times \;100$$
where Abs_control_ = the absorbance of control. Abs_sample_ = the absorbance of sample.

EC_50_ value, which was defined as the substrate concentration (mg mL^−1^) needed to inhibit 50% of the DPPH radical scavenging activity, was employed to express the results. EC_50_ was determined from a graph of antioxidant activity (%) versus extract concentration of each pear (mg mL^−1^ sample).

### 2.6. Statistical Analysis

The significance of the pre-treatments (ultrasound and microwave) and drying process (i.e., temperature) effects were assessed employing a one-way analysis of variance (ANOVA). The significance level assumed in all situations was fixed at *p* < 0.05. Moreover, Pearson’s correlation tests were performed to determine the relationship between the white index (WI) and total phenolic (TPC), the white index and antioxidant activity (EC_50_) and total phenolic content and antioxidant activity. Concerning the modeling approaches, the 95% standard error of the parameter (SE) and statistical indicators of the quality of the regression (coefficient for determination (R^2^) and standard deviation of the experimental error (s)) were also calculated [58]. All statistical analyses were carried out using IBM SPSS^®^ Statistics^®^ 24 for Windows (SPSS Inc., Chicago, IL, USA).

## 3. Results and Discussion

### 3.1. Color Evaluation

Color parameters of fresh pears and control (C), microwave (MW) and ultrasound (US) pre-treated dried samples at 50, 55 and 60 °C are presented in Table 1, and include the L*, the whiteness index (WI), and total color difference (ΔE). According to the results, color parameters were affected by both applied pre-treatments and drying temperatures. After the drying process, the L* value decreased in all microwave-treated samples at each drying temperature, and the lowest *L** value was in microwave-treated dried samples at 50 °C. It was visually observed that all microwave treated pears’ color got browner after hot air drying compared to control and ultrasound treated dried ones. Similar observations were reported by Krokida et al. [59] wherein the lightness parameter (L*) decreased significantly in microwave treated apple, potato, banana and carrot samples conventionally dried at 70 °C. A similarity of L* values between the control and ultrasound pre-treated samples dried at 50 and 55 °C was observed. The highest L* value was observed in ultrasound pre-treated pears dried at 60 °C as 79.05 ± 0.25, while the L* values of fresh samples were 78.60 ± 0.25. The highest L* values of ultrasound pre-treated pears dried at 60 °C indicated that the ultrasound application with a higher drying temperature of 60 °C was sufficient to preserve the fresh fruit’s lightness. Thus, the pear’s original color could be better protected when the samples received the combination of ultrasound pre-treatment and higher drying temperatures of 60 °C.

Concerning the whiteness index, the drying process reduced its value for all dried samples. In particular, a significant decrease was observed in all microwave-treated dried pears. However, ultrasound-pre-treated dried pears at 60 °C demonstrated similar whiteness index values to those of fresh ones (*p* > 0.05).

The total color difference is a crucial parameter for dried fruits and vegetables, being an indicator of the human eye’s ability to differentiate between products [60]. The drying temperature also influenced the values of total color changes (∆E), revealing a decreasing value with the increase of temperature from 50 to 60 °C. These results showed the color deterioration during the drying process being more pronounced when the lowest drying temperature was employed. This tendency may be explained by the browning reactions or the formation of browning products occurring at lower-temperature and long-time exposure to the drying process at a low temperature (50 °C). The highest values of ∆E were observed for all microwave-pre-treated dried pears. These highest Δ*E*s were likely due to high differences in the L* (lightness) values. This trend may be attributed to non-uniform temperature distribution during microwave treatment such that few regions of pear samples could get heated very rapidly, while the remaining region could get heated to a lesser extent. Controlling heating uniformity is an important parameter for obtaining high quality microwave-dried fruits. During microwave applications, the constant microwave power can cause an increase in the average product temperature, and overheating of the food material can be prevented by controlling the microwave power [61]. A large number of factors may influence temperature distribution during microwave applications such as the thickness, geometry and dielectric properties of foods and microwave energy [62]. The highest color alterations of microwave-treated dried samples may be dependent on the thickness of pear, microwave time or loss of homogeneity during the microwave treatment.

The lowest value of ∆E was observed in ultrasound-pre-treated pears dried at 60 °C, indicating that the combined pre-treatment of the ultrasound application and higher drying temperature better helped to preserve the original color of pear in dried snacks (Figure 1). At this point, it could be concluded that ultrasound pre-treatment could be applied to control or inhibit browning reactions and attain the desirable color of final dried pears [63].

### 3.2. Total Phenolic Content (TPC)

The total phenolic content of fresh pears and control (C), microwave (MW)- and ultrasound (US)-pre-treated pears dried at 50, 55 and 60 °C is summarized in Figure 2. The present experiments demonstrated that drying temperature and microwave and ultrasound pre-treatments had a remarkable effect on pear samples’ total phenolic content. The total phenolic content of fresh pear slabs was found to be 336.82 mg GAE/100 g d.m. Total phenolic content levels generally decreased after the drying process. A significant reduction of total phenolic content was observed in all microwave-treated pears after drying, and there were no statistical differences (*p* > 0.05) between all microwave pre-treated dried samples. These results indicated that microwave pre-treatment had a negative impact on total phenolic content and damaged the nutritional composition of all dried pears. Moreover, microwave-treated dried pears’ total phenolic content was slightly lower than control and ultrasound-treated dried pears (*p* < 0.05). The phenolic compounds’ degradation by microwave treatment might be due to the heating effect caused by electromagnetic radiation during the microwave application.

The most significant changes in microwave-treated dried pears’ total phenolic composition might be attributed to the structural changes in fruits due to the combined drying method (MW application and hot air drying). During the microwave treatments, an observed shrinkage phenomenon could cause internal stress (non-uniform temperature) and surface tension, resulting in surface microcracks and leakage of exudate from the pears. Contrary to our findings, Hayat et al. [64] noticed that microwave energy could increase the bioavailability of some phenolic compounds (i.e., free phenolic compounds) by liberating them from the food matrix. Microwave heating is associated with an essential phenomenon of selective heating. The release of phenolic compounds is believed to be attributed to the selective heating of certain phenolic compounds in the microwave field or physical forces between the food matrix and phenolic compounds. The thermal instability of microwave heating could be one of the most important reasons for phenolic compound losses. Furthermore, the thermal instability of phenolic compounds in the food matrix can play an important role in the separation process [65]. However, not much knowledge about the impacts of microwave application on quality attributes of hot-air-dried fruits is currently available.

Concerning ultrasound pre-treatment, no significant differences (*p* > 0.05) were observed between all dried control pears and ultrasound-pre-treated ones up to 60 °C. Among all dried pears, the highest total phenolic content was 345.60 mg GEA/100 g d.m. in ultrasound pre-treated pears dried at 60 °C, and only this value was not statistically different from the fresh pears. Based on these results, the combined application of ultrasound pre-treatment and higher drying temperature of 60 °C affected the pears’ total phenolic content positively. This behavior could be explained by the higher processing temperature and less exposure time, which contributed to a protective effect against oxidative and heat damage to pears’ phenolic composition. Such situations may be attributed, in addition, to the better availability and extractability of antioxidant compounds, which can be enhanced by larger pores in the pear tissue, by an ultrasound application, thereby improving the extraction of polyphenols sample preparation [66,67].

The decrease of total phenolic compounds due to the drying process may be associated with the modifications in phenolic compounds’ chemical structure or the binding of polyphenols with other compounds, such as proteins [68,69].

In this case, the ultrasound application with higher drying temperature (60 °C) exhibited better retention of pear samples’ antioxidant activity.

### 3.3. DPPH Radical Scavenging Activity

The antioxidant activity of fresh and dried pears with different pre-treatments was evaluated by a DPPH radical scavenging activity assay (please see Figure 3). The lowest EC_50_ corresponds to the highest pears’ antioxidant activity. The fresh samples provided a radical scavenging activity of 9.39 mg/mL. After the drying process, the samples had less antioxidant activity except for ultrasound-pre-treated samples dried at 60 °C. According to these results, dried pears’ higher antioxidant activity value was positively correlated with the higher drying temperature. In this case, the lower drying temperature produced more degradation of pears’ antioxidant activity. Previous studies stated that the lowest antioxidant activity of conventionally air-dried fruits and vegetables was linked to lower processing temperature and longer exposure time to heat [27,70,71]. Also, Nicoli et al. [72] stated that some fruits’ antioxidant capacity could be maintained or enhanced by the formation of new antioxidant compounds. According to Kamiloglu and Capanoglu [73] and Vega-Gálvez et al. [71], generation and accumulation of Maillard-derived melanoidins, having a different degree of antioxidant activity, could also increase the antioxidant activity of fruits.

The microwave treatment resulted in significantly less antioxidant activity in dried pears than control and ultrasound-pre-treated dried pears, demonstrating that some phenolic acids were probably degraded by microwave application. The antioxidant activity values of control and ultrasound-pre-treated dried pears were significantly different (*p* < 0.05) than microwave-treated dried ones. The significantly highest values of EC50 (the lowest antioxidant activity) were obtained in “Rocha” pears subjected to microwave pre-treatment drying at 50 °C. This behavior could be due to the combined application of microwave pre-treatment and drying process conditions of exposure time and temperature: lower temperature and longer exposure time produced more degradation of polyphenols and consequently, the loss of antioxidant activity. Moreover, microwave pre-treatment probably caused significant structural changes in pear cells, which led to the loss of antioxidant activity on the fruit surface.

All pear samples obtained from ultrasound pre-treatment exhibited better antioxidant properties. However, no statistical differences (*p* > 0.05) were observed between ultrasound-treated pears dried at 55 °C and control ones dried at 60 °C. The ultrasound-treated pears at a higher drying temperature of 60 °C showed the highest antioxidant activity retention. Such a situation could probably be attributed to the protection of pear cellular structure degradation by combined application (ultrasound treatment and higher drying temperature of 60 °C), which may have improved the samples’ antioxidant activity.

Based on these results, microwave and ultrasound pre-treatments differentiated the phenolic contents and antioxidant potential in dried pears. Nevertheless, numerous factors may affect food products’ antioxidant capacity, such as different drying methods, temperatures, applied pre-treatments, antioxidant assays, main antioxidant compounds’ chemical structures and the interactions of several antioxidant reactions [55,70].

In our case, this proper combined drying method (ultrasound treatment and higher drying temperature of 60 °C) may be efficient, preserving the antioxidant activity and phenolic compounds in “Rocha” pear samples.

Pearson’s correlation test was applied to determine the correlation between the white index (WI) and total phenolic content as well as the antioxidant activity (EC_50_ value). The total phenolic content and antioxidant activity influenced the white index (WI) of “Rocha” pears. Total phenolic content showed a high positive correlation with the white index (*r* = 0.928). At the same time, the antioxidant activity (EC_50_) had a high negative correlation with the white index (*r* = −0.936) because a lower EC_50_ indicated higher antioxidant activity. If r is regarded as an absolute value, it is possible to observe a high correlation.

Moreover, it was confirmed herein by statistically significant negative Pearson’s correlation among the total phenolic content and EC_50_ value (*r* = −0.845). This result indicated that with a higher polyphenol content, a lower amount of “Rocha” pear tissue was required to scavenge 50% of initial free radical concentration (lower EC_50_ value). The analogous relationship was reported by Rahman et al. [74] and Wiktor et al. [67].

### 3.4. Drying Kinetics: Experiments and Empirical Models

Figure 4a–c presents the drying curves of control, microwave and ultrasound pre-treated pear slices at the investigated temperatures of 50, 55 and 60 °C. The drying curves depended on the applied pre-treatments (i.e., microwave and ultrasound) and drying conditions (i.e., temperature and time). Fresh pears were characterized by moisture content of 6.42 ± 0.66 kg water/kg d.m. and water activity of 0.98 ± 0.004. The pear drying process was performed to a final moisture content of less than 0.1 ± 0.01 kg water/kg d.m. and average water activity of 0.41 ± 0.03 (<0.05).

No constant rate period was detected in all samples within the studied drying experimental conditions. Only the falling rate period was observed in all samples at each drying temperature, indicating that the tray dryer’s drying process was controlled by internal mass transfer diffusion (moisture migration within pear samples). Similar drying characteristics have been observed in several fruits’ drying, such as the persimmon [75], apple [52], grape [38] and pineapple [76].

All pear samples’ drying curves presented similar behavior: moisture content decreased and the rate of moisture loss decreased as time progressed. Moreover, the drying process was accelerated when the temperature increased from 50 to 60 °C. Drying behavior with increasing drying temperature was explained by Nascimento et al. [77] and Tao et al. [78]. The temperature rising reduces the relative humidity of the air, increases the moisture gradient between food materials and air and promotes the moisture movement within foods, thus enhancing drying time, but may cause also case-hardening. If case-hardening occurs, the diffusion path of moisture from the internal layers toward the product surface is impeded. As a consequence, diffusion will become slower, and the drying time will be prolonged. A similar phenomenon has been observed for yam slices’ convective drying [79].

As shown in Figure 4, in microwave-pre-treated (539 W for 4 min) pear slices, the moisture content decreased faster than in control or ultrasound-pre-treated (US) samples, thus enhancing the drying process. In this way, the microwave pre-treated samples had shorter drying times at all investigated temperatures. Pear samples were dried to a final moisture content less than 0.1 kg water/kg d.m. Considering the practical application, the smaller time required to achieve the same final moisture content was 315 min for microwave pre-treated and drying at 60 °C. Corresponding values for control and ultrasound samples were 453 min and 492 min, respectively. Similar drying behavior was observed at drying temperatures of 50 and 55 °C. This behavior may be explained by the internal (volumetric) heating caused by moisture migration to the pear surface during microwave pre-treatment. Less drying time in the microwave and its combined applications may be associated with the rapid mass transfer within food materials during microwave heating. Heat was generated within the food due to the absorption of microwave energy, and it creates high internal pressure, temperature and concentration gradients. Therefore, the flow rate of the liquid through the food to the boundary is increased.

Concerning ultrasound-prior drying applications, at a temperature of 50 °C, samples had a shorter drying time than control ones. The lower drying temperature and longer exposure time may have produced higher damage to control samples. However, there was no drying time reduction in ultrasound-treated pears dried at 55 and 60 °C in comparison with control samples at the same temperatures. Under these conditions, the drying time of control and ultrasound pre-treated samples were quite similar. This could probably be caused by either the fact that ultrasound pre-treatments were applied through the vacuum packaging and not by the samples’ immersion in the water medium or by structural changes of pear tissue that occurred during ultrasound pre-treatments. The structural properties of the food material are decisive for the ultrasound waves’ action [77]. From this point of view, the pears’ composition and structure may be modified by ultrasound application and can affect pears’ ability to be dried.

The ultrasound application also causes a rapid alternative compression and expansion of the food matrix called a sponge effect. This mechanism is a direct effect of ultrasound, leading to the creation of microchannels and facilitating intracellular liquid removal to the surroundings. Another phenomenon associated with ultrasonic waves is the formation of thousands of cavitation bubbles in liquids. In our case, considering the effect of ultrasound application on “Rocha” pears ‘drying process, these obtained findings could be associated with the pear samples’ isolation by vacuum-packaging from the liquid medium (water) during the ultrasound application. This type of pre-treatment procedure probably caused a limited effect of cavitation, and only sponge effect occurred. Therefore, it can be noted that the occurrence of this sponge effect was insufficient to accelerate the drying process and moisture movement. These observations are in good agreement with results reported for ultrasound-pre-treated dried carrot slices [24]. On the contrary, in the cases where pre-treatment with ultrasound was applied before drying and food materials were directly immersed in a liquid medium, a significant reduction of processing time was observed for pineapple [42], melon [21] and mulberry [20].

“Rocha” pears’ drying characteristics were affected by many factors such as the fruit’s structure, applied pre-treatments, and drying temperature and time. From this viewpoint, obtaining an in-depth understanding and description of the microwave and ultrasound effects, and the involved mechanisms may also improve the drying process. Microwave application to the drying of pears had a remarkable impact on reducing process time, which may be attributed to the two microwave mechanisms of ionic polarization and dipole rotation. In ionic polarization, an electric field is applied to make ions move and collide with each other. As a consequence, their kinetic energy is converted into heat inside the food materials [80]. Both mechanisms are related to the “volumetric heating” of microwave energy. The ultrasound pre-treatment did not accelerate the drying process of pear fruits. Therefore, detailed information on ultrasound mechanisms (i.e., sponge effect and cavitation) and their effects on the drying process are needed.

In order to predict the moisture content (X) as a function of drying time (min), empirical models (presented in Section 2.4) were fitted to the experimental data. The drying kinetic parameters for each attempted model are reported in Table 2, where k is the drying rate constant (min^−1^ or min^−N^), and a and N are drying parameters. The corresponding statistical parameters are summarized in Table 3. The coefficient of determination (R^2^) and the standard deviation of the experimental error (s) were the primary criteria to select the best model to account for variation in the drying curves of pear samples. A good fitting corresponded to the highest R^2^ value and the lowest value of s [47].

The statistical parameter estimations showed that R^2^ and s values ranged, respectively, from 0.974 to 1.000 and from 0.0255 to 0.2038. According to model parameters in Table 2, for the Newton and Henderson & Pabis models, the drying rate k had values of 0.446 × 10^−2^ to 0.925 × 10^−2^ min^−1^. The Page model gave the best results and demonstrated good agreement with experimental data (Figure 4), presenting the highest R^2^ and the lowest s values among all tested empirical models. Hence, the Page model could predict with sufficient accuracy the evolution of moisture content for “Rocha” pear slabs at each drying temperature. Similar observations were reported by Doymaz and İsmail [37] and Senadeera et al. [40].

## 4. Conclusions

The influences of air drying temperature (50, 55 and 60 °C) and innovative pre-treatments (i.e., microwave and ultrasound applications) on quality attributes and drying behaviour of “Rocha” pear slices were investigated. Drying process conditions of pears significantly influenced all quality descriptors: lower drying temperature and longer exposure time resulted in more quality deterioration. Moreover, ultrasound pre-treatment combined with higher drying temperature (60 °C) improved the quality attributes of pear samples: better color preservation and retention of nutritional compounds (i.e., total phenolic content and antioxidant activity). On the other hand, microwave pre-treatment had a negative impact on the overall quality of dried pears.

Regarding the characteristic drying curve, only the falling rate was observed at all investigated conditions. The experimental results evidenced that the drying time decreased with increasing drying temperatures. Microwave pre-treatment was more effective in intensifying the drying process due probably to the volumetric heating of vapors generated inside the pear material. Therefore, microwave pre-treated pears had shorter drying times in comparison with control and ultrasound-pre-treated slices. On the other hand, ultrasound pre-treatment did not accelerate the drying process, which could be explained by the different ultrasound application mechanisms resulting probably from the following effects: the limited cavitation phenomenon and only the sponge effect occurring in vacuum-packaged pear samples. Among the applied empirical models, the Page model provided the best fit results (with the highest correlation factor, R^2^, and the lowest standard deviation of experimental error, s) to describe the drying behavior of “Rocha” pears at each drying temperature accurately.

It can be concluded that each pre-treatment exhibits a different effect mechanism on the drying characteristics and the preservation of nutritional and quality parameters. Despite the ultrasound not accelerating the drying process of pear fruits, quality attributes of dried pears were better retained. Current approaches to understanding and improving drying process efficiency state that more research is needed regarding ultrasound application conditions (i.e., time, power, etc.) and the ultrasound mechanism. In this context, ultrasound pre-treatment may be a promising technique to obtain high-quality dried fruits, being an alternative to traditional pre-treatments and suitable to an industrial context. Additional studies must be carried out to determine the influence of pre-treatments’ mechanisms on the dried product’s final quality.

Future research will be devoted to investigating the description and characterization of “Rocha” pear rehydration behavior. The selection of appropriate pre-treatments seems crucial for optimizing rehydration process conditions and new product design.

## Figures and Tables

**Figure 1 foods-10-00853-f001:**
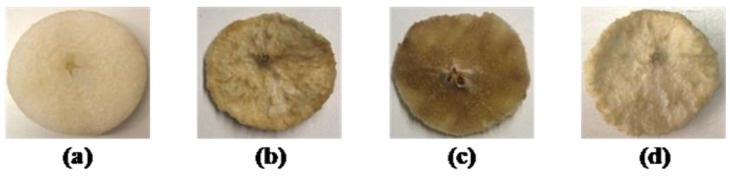
“Rocha” pear slabs: fresh (**a**), control (**b**), microwave treated (**c**) and ultrasound treated (**d**) dried at 60 °C.

**Figure 2 foods-10-00853-f002:**
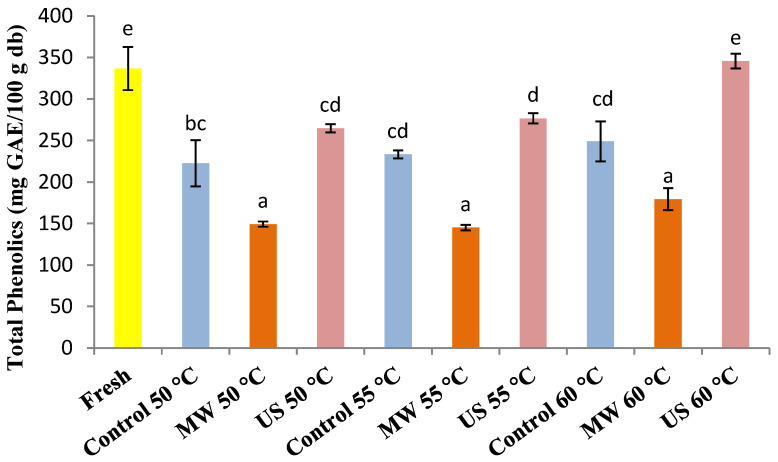
Total phenolic content of fresh and control (C), microwave (MW) and ultrasound (US) treated “Rocha” pears dried at 50, 55 and 60 °C. Values with the same letter were not significantly different (*p* > 0.05).

**Figure 3 foods-10-00853-f003:**
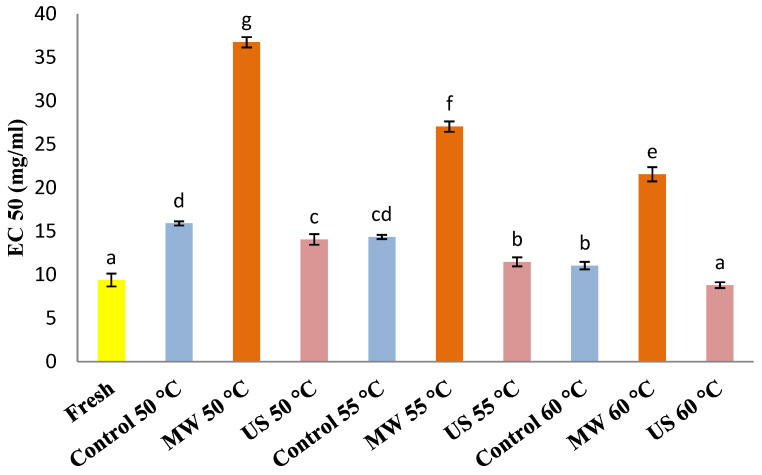
Antioxidant activity of fresh and control (C), microwave (MW) and ultrasound (US) treated “Rocha” pears dried at 50, 55 and 60 °C. Values with the same letter were not significantly different (*p* > 0.05).

**Figure 4 foods-10-00853-f004:**
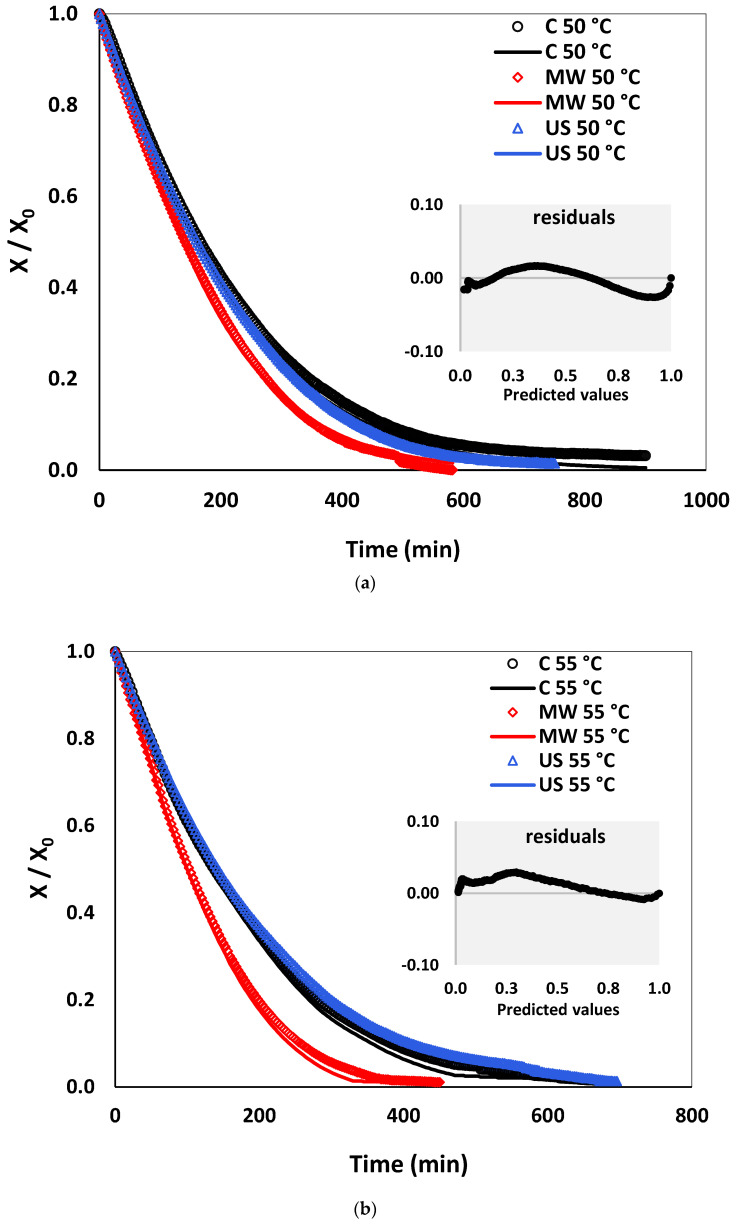
Experimental (symbols) and Page model-predicted (lines) drying curves of control and pre-treated (microwave—MW and ultrasound—US) samples at (**a**) 50 °C, (**b**) 55 °C and (**c**) 60 °C.

**Table 1 foods-10-00853-t001:** Color parameters of control (C), microwave (MW) and ultrasound (US) pre-treated “Rocha” pears dried at 50, 55 and 60 °C.

Sample	L*	WI	∆E
Fresh	78.60 ± 0.90 ^e^	72.11 ± 1.00 ^f^	-
C50 °C	71.35 ± 1.98 ^c^	61.21 ± 1.21 ^c^	11.57 ± 0.71 ^d^
MW50 °C	55.77 ± 2.99 ^a^	48.35 ± 1.17 ^a^	23.41 ± 2.18 ^f^
US50 °C	73.77 ± 0.89 ^cd^	63.23 ± 0.88 ^cd^	9.80 ± 0.71 ^cd^
C55 °C	72.46 ± 1.66 ^c^	63.03 ± 0.60 ^cd^	10.23 ± 0.78 ^cd^
MW55 °C	58.35 ± 0.84 ^ab^	51.38 ± 1.15 ^b^	21.99 ± 0.92 ^f^
US55 °C	77.15 ± 0.42 ^de^	68.00 ± 0.48 ^e^	6.11 ± 0.32 ^ab^
C60 °C	75.06 ± 0.80 ^cde^	64.96 ± 0.67 ^d^	8.13 ± 0.71 ^bc^
MW60 °C	60.53 ± 0.68 ^b^	52.88 ± 0.95 ^b^	14.91 ± 0.81 ^e^
US60 °C	79.05 ± 0.25 ^e^	70.96 ± 0.21 ^f^	3.86 ± 0.23 ^a^

Values in the same column with the same letter were not significantly different (*p* > 0.05).

**Table 2 foods-10-00853-t002:** Drying kinetics model parameters of “Rocha” pears dried at 50, 55 and 60 °C.

Model Name	Temperature	Parameters	Control	Microwaved	Ultrasound
Newton	50 °C	k × 10^2^ (1/min)	0.446 ± 0.005	0.558 ± 0.024	0.478 ± 0.009
	55 °C	k × 10^2^ (1/min)	0.578 ± 0.015	0.808 ± 0.043	0.546 ± 0.023
	60 °C	k × 10^2^ (1/min)	0.604 ± 0.013	0.851 ± 0.071	0.608 ± 0.014
Henderson & Pabis	50 °C	a	1.061 ± 0.007	1.074 ± 0.019	1.068 ± 0.010
		k × 10^2^ (1/min)	0.481 ± 0.006	0.598 ± 0.026	0.508 ±0.010
	55 °C	a	1.068 ± 0.012	1.098 ± 0.025	1.058 ± 0.010
		k × 10^2^ (1/min)	0.615 ± 0.01	0.881 ± 0.046	0.577 ± 0.012
	60 °C	a	1.074 ± 0.018	1.090 ± 0.077	0.632 ± 0.017
		k × 10^2^ (1/min)	0.646 ± 0.026	0.925 ± 0.031	1.041 ± 0.011
Page	50 °C	k × 10^2^ (1/min^N^)	0.169 ± 0.004	0.154 ± 0.015	0.165 ± 0.008
		N	1.184 ± 0.005	1.240 ± 0.021	1.191 ± 0.009
	55 °C	k × 10^2^ (1/min^N^)	0.213 ± 0.014	0.170 ± 0.018	0.282 ± 0.029
		N	1.186 ± 0.013	1.310 ± 0.023	1.124 ± 0.022
	60 °C	k × 10^2^ (1/min^N^)	0.184 ± 0.062	0.177 ± 0.029	0.312 ± 0.019
		N	1.224 ± 0.019	1.319 ± 0.031	1.126 ± 0.01

**Table 3 foods-10-00853-t003:** Correlation coefficients (R^2^ and s) of empirical drying models.

Model Name	Temperature	Correlation Coefficients	Control	Microwaved	Ultrasound
Newton	50 °C	R^2^	0.995	0.984	0.991
		s	0.1168	0.1734	0.1622
	55 °C	R^2^	0.991	0.978	0.993
		s	0.1477	0.1925	0.1321
	60 °C	R^2^	0.986	0.974	0.994
		s	0.2038	0.1999	0.1282
Henderson & Pabis	50 °C	R^2^	0.997	0.989	0.995
		s	0.0835	0.1432	0.1263
	55 °C	R^2^	0.994	0.986	0.996
		s	0.1202	0.153	0.1001
	60 °C	R^2^	0.991	0.983	0.996
		s	0.1612	0.1733	0.1071
Page	50 °C	R^2^	1	0.997	0.999
		s	0.0255	0.0595	0.0481
	55 °C	R^2^	0.999	0.998	0.998
		s	0.0541	0.0534	0.0776
	60 °C	R^2^	0.999	0.997	0.999
		s	0.065	0.0737	0.0499

## Data Availability

The datasets generated for this study are available on request to the corresponding author.

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
