# Peer review of "Microwave and Ultrasound Pre-Treatments for Drying of the “Rocha” Pear: Impact on Phytochemical Parameters, Color Changes and Drying Kinetics"

_foods, 2021, doi:10.3390/foods10040853_

Round 1
Reviewer 1 Report
Development of methods to improve the dehydration efficiency of biomaterials has high relevance in the field of food science and engineering. Considering the effects of microwave irradiation and ultrasound on physicochemical structure of food and food raw materials, these pretreatment processes are considered as suitable to improve the energy efficiency of drying and the quality and sensory properties of food.
Therefore, manuscript foods-1134446 has an interesting topic for the science and practice, as well. Beside the drying kinetics, authors investigated the change of colour, total phenolic content and antioxidant activity of pear.
The manuscript is generally well written with a logic structure. Introduction summarized well the theoretical background of the research. The novelty of the research and research motivations are well defined. Materials and methods are clearly given. Applied methods can be considered as adequate to investigate the research questions. Manuscript contains significant and valuable results that are discussed with relevant references.
Comments and suggestions:
It is not clearly given why use the authors the frequency and time of ultrasound pretreatments.
How was measured and controlled the temperature during microwave treatments?
In my opinion, it would be more logical to discuss firstly the drying kinetic results and the change of quality parameters after.
I suggest the authors to checky the typos in the manuscript (‘samplesobtained’ in line 359, ’ showedthe’ in line 352, ’ curvecharacteristic,’ in linw 493, for instance)
In my opinion, the reason for the highest EC50 by MW50C is not discussed in details.
Authors concluded in section 4 ‘ultrasound pre-treatment did not accelerate the drying process, which can be explained by the different ultrasound application mechanisms resulting probably from the following effects: the limited cavitation phenomenon and only sponge effect occurred in vacuum501 packaged pear samples’ (line 499-501) and one paragraph later that ‘ultrasound pre-treatment may be a promising technique to obtain high quality dried fruits..’ (line 505-507). I suggest the authors to rephrase the Conclusion section to avoid the misunderstanding sentences.
Author Response
Please find in the attached file the detailed answer to Reviewer 1 comments and suggestions. We believe the manuscript is now improved and that can now be accepted, We thank you for the contribution of Reviewer 1.

Reviewer 2 Report
-some interesting results are shown
- some important information is missing:
- microwave treatment:
- what is the fresh mass of the pear slabs, what that after MW heating? (What was the energy per mass? How much water is evaporated?)
- how were the slabs arranged, was a rotating plate used?
- was the temperature of the slabs measured?
- drying:
- where is the air speed measured (at the surface of the slabs)
- was the drying air circulated?
- ultrasound treatment:
- what was the actual power used (only the range is stated)
- comments to different paragraphs:
- abstract:
- mw power without stating mass is not useful
- introduction:
- ll.43-54: a dried pear slab is a completely different product than a fresh pear, so the arguments for „new strategies for improving the quality of pears after harvest“ are nor really sound. Besides drying is by far not a new strategy
- ll. 62-70: the phenomena, stated as „furthermore“ by the authors are the reasons (and not the consequences) for the before mentioned effects.
- ll.71-79: the microwave technology is presented in a too simplified and general way, so the mw technology does not penetrate the food but only the microwaves, they do not heat generally „from inside to outside“, this is depenedent on food properties (permittivity, water content, …), geometry (curvature, arrangement in oven…) and so on. Also the drawbacks cannot be stated in such a general manner.
- Materials and methods
- l. 116: cavitation may also take place within food (when water is present)!
- l. 122: what was the actual power used (only the range is stated)?
- ll. 127-130: what is the fresh mass of the pear slabs, what that after MW heating? (What was the energy per mass? How much water is evaporated?) how were the slabs arranged, was a rotating plate used? was the temperature of the slabs measured?
- l.140: what means „constant mass“ (within what range and within what time?)
- ll.145-146: was the drying air circulated and was the humidity constant?
- Ll.154-155: How is the equation simplified? Is Xe a paramter in fitting or is it set?
- Table 1: the model of Henderson and Pabis is not really justified (the parameter a is meaningless (it has to be close to 1)
- l.161: the unit of k is in the model of Page not (1/min) but (1/minN)!
- Results and Discussion
- Ll.314-320: In a well.mixed product (with low amounts of compounds beside water) selective heating oft he compounds is rather improbable, thernal instability will be the reason for losses.
- Ll.394-398: How did the authors try to detect a constant rate period? Please show the drying rates (over the water loading) to support this statement!
- Ll.451-456: the 2 stated mw mechanisms lead both to heating!
- Table 3: as stated above : the unit of k in the model of Page is not (1/min) but (1/minN)! Since N differs between the experiments k cannot be compared!
Author Response
Please find in the attached file the detailed answer to Reviewer 2 comments and suggestions. We believe the manuscript is now improved and that can now be accepted, We thank you for the contribution of Reviewer 2.

Round 2
Reviewer 1 Report
Development of methods for enhanced drying efficiency, and improve the drying kinetics for different biomaterials has high relevance for the science and practice, as well. Therefore, manuscript foods-1134446 has a topic that can be considered as novel and interesting for the readers.
Authors have revised the manuscript thoroughly according to reviewers’ comments and suggestions. The manuscript is well written with a logic structure.
During revision, authors give the power and frequency of US apparatus, and references have been given which provide the information for the selection of method and treatment parameters. Materials and methods section has been amended by more detailed information and data.
Typos have been corrected in the manuscript. Discussion section has been amended and rephrased to make it more clear and complete.
I agreed all modifications made by the authors and accept all answers.
After the revision the overall scientific quality of manuscript improved significantly and is has been more clear and complete in general.